# Distribution Embedding Networks for Generalization from a Diverse Set of Classification Tasks

**Lang Liu**                                                                   *liu16@uw.edu*
*University of Washington*

**Mahdi Milani Fard**                                              *mmilanifard@google.com*
*Google Research*

**Sen Zhao**                                                            *senzhao@google.com*
*Google Research*

**Reviewed on OpenReview:** *https://openreview.net/forum?id=F2rG2CXsg0*

## Abstract

We propose Distribution Embedding Networks (DEN) for classification with small data. In the same spirit of meta-learning, DEN learns from a diverse set of training tasks with the goal to generalize to unseen target tasks. Unlike existing approaches which require the inputs of training and target tasks to have the same dimension with possibly similar distributions, DEN allows training and target tasks to live in heterogeneous input spaces. This is especially useful for tabular-data tasks where labeled data from related tasks are scarce. DEN uses a three-block architecture: a covariate transformation block followed by a distribution embedding block and then a classification block. We provide theoretical insights to show that this architecture allows the embedding and classification blocks to be fixed after pre-training on a diverse set of tasks; only the covariate transformation block with relatively few parameters needs to be fine-tuned for each new task. To facilitate training, we also propose an approach to synthesize binary classification tasks, and demonstrate that DEN outperforms existing methods in a number of synthetic and real tasks in numerical studies.

## 1 Introduction

While machine learning has made substantial progress in many technological and scientific applications, its success often relies heavily on large-scale data. However, in many real-world problems, it is costly or even impossible to collect large training sets. For example, in online spam detection, at any time, we may only possess dozens of freshly labeled spam results. In health sciences, we may only have clinical outcomes on a few hundred study subjects. Few-shot learning (FSL) has recently been proposed as a new framework to tackle such small data problems. It has now gained huge attention in many applications such as image classification (Koch et al., 2015; Finn et al., 2017), sentence completion (Vinyals et al., 2016; Munkhdalai et al., 2018), and drug discovery (Altae-Tran et al., 2017); see Wang et al. (2020) for a survey.

The core idea behind most of FSL methods is to augment the limited training data with prior knowledge, e.g., images of other classes in image classification or similar molecules' assays in drug discovery. In *meta-learning* based FSL, such prior knowledge is formulated as a set of related training tasks assumed to follow the same task distribution (Finn et al., 2017), with the goal that the trained model could quickly adapt to new tasks. In practice, however, given an arbitrary target task, the degree and nature of its relationship to the available auxiliary training data is often unknown. In this scenario, it is unclear if existing approaches can extract useful information from the training data and improve the performance on the target task. In fact, there is empirical evidence suggesting that learning from unrelated training tasks can lead to negative adaptation (Deleu & Bengio, 2018). In our numerical studies, we also observe similar behavior; existing FSL approaches perform poorly when training tasks are unrelated to the target task.

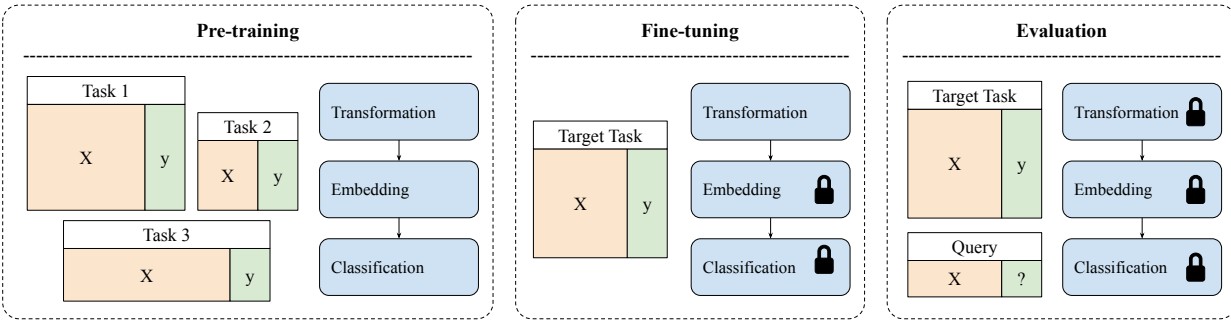

Figure 1: Training and evaluation of DEN. We first pre-train DEN on training tasks with heterogeneous covariate spaces. We then fine-tune the transformation block on a few labeled examples from the target task, and use the fine-tuned model for classification on the query set.

In this paper, we investigate this important but under-studied setting—few-shot meta-learning with possibly unrelated tasks. We specifically focus on classification tasks with *tabular* data. Unlike machine learning with image and text inputs, large datasets of related tasks are not available for generic tabular-data classification.

A key challenge in this setting is that the input, in the form of covariate vectors for training and target tasks, can live in different spaces and follow different distributions with possibly different dimensions. Existing meta-learning techniques often assume a *homogeneous* input space across tasks and thus cannot be directly applied in such cases with *heterogeneous* covariate spaces.

In this work, we propose Distribution Embedding Networks (DEN) for meta-learning on classification tasks with potentially *heterogeneous* covariate spaces. DEN consists of a novel three-block architecture. It first calibrates the raw covariates via a transformation block. A distribution embedding block is then applied to form an embedding vector serving as the "summary" of the target task. Finally, a classification block uses this task embedding vector along with the transformed query features to form a prediction.

Since the tasks can be unrelated, we learn a different transformation block for each task to form task-invariant covariates for the rest of the network. In other words, the transformation block is *task-dependent*. We keep the *task-independent* embedding and classification blocks fixed after *pre-training*, and use a few labeled examples from the target task (i.e., the support set) to *fine-tune* the task-dependent transformation block. Since our setting is significantly more challenging than the standard few-shot meta-learning setting due to the heterogeneity among training and target tasks, we assume that we have access to a slightly larger support set compared to the FSL setting (e.g., 50 examples in total across all classes rather than 5 examples per class). We further assume that the support set follows the same distribution as the query set. To address the challenge of variable-length covariates, the classification block is built upon a Deep Sets architecture (Zaheer et al., 2017). Figure 1 shows an overview of the architecture and training and evaluation mechanisms.

To summarize our main contributions: (I) We propose a method for meta-learning with possibly unrelated tabular-data training tasks—an important setting that expands the application of meta-learning but has rarely been investigated in the literature; (II) We propose the three-block architecture, allowing the model to be pre-trained on a large variety of tasks, and then fine-tuned on an unrelated target task; we provide a scenario in which our three-block architecture can perform well; (III) Described in Section 5, we design a procedure to generate artificial tasks for pre-training, and empirically verify its effectiveness when testing on real tasks. This provides a principled way to generate training tasks and alleviates the cost of collecting real training tasks. (IV) We compare DEN with various existing FSL approaches on both simulated and real tasks, showing improved performance in most of the tasks we consider.

## 2 Related Work

There are multiple generic techniques applied to the meta-learning problem in the literature (Wang et al., 2020). The first camp learns similarities between pairs of examples (Koch et al., 2015; Vinyals et al.,

2016; Bertinetto et al., 2016; Snell et al., 2017; Oreshkin et al., 2018; Wang et al., 2018; Sung et al., 2018; Satorras & Estrach, 2018; Liu et al., 2019; Mishra et al., 2018). For an unlabeled example on a new task, we use its similarity score with labeled examples of the given task for classification. The second camp of optimization-based meta-learning aims to find a good starting point model such that it can quickly adapt to new tasks with a small number of labeled examples from the new task. This camp includes different variants of MAML (Finn et al., 2017; Lee & Choi, 2018; Finn et al., 2018; Grant et al., 2018; Rusu et al., 2019) and Meta-Learner LSTM (Ravi & Larochelle, 2017). More recently, Lee et al. (2020; 2021) proposed to learn task-specific parameters for the loss weight and learning rate for out-of-distribution tasks. Their use of task-embedding is conceptually similar to DEN. The third camp is conceptually similar to topic modeling, such as Neural Statistician (Edwards & Storkey, 2017) and CNP (Garnelo et al., 2018), which learn a task specific (latent) embedding for classification. The final camp utilizes memory (Santoro et al., 2016; Kaiser et al., 2017; Munkhdalai & Yu, 2017; Munkhdalai et al., 2018). Note that all the above methods assume that all training and target tasks are related and share the same input space.

A closely related problem is Domain Generalization (DG) which estimates a functional relationship between the input $\boldsymbol{x}$ and output $y$ given data from different domains (i.e., with different marginal $P(\boldsymbol{x})$); see, e.g., Wang et al. (2021) and Zhou et al. (2021) for surveys. The core idea lies behind a large class of DG methods is to learn a domain-invariant feature representation (see, e.g., Muandet et al., 2013; Li et al., 2018a;b; Shankar et al., 2018; Shen et al., 2018), which aligns the marginal $P(\boldsymbol{x})$ and/or the conditional $P(y|\boldsymbol{x})$ distributions across multiple domains. In a similar spirit, DEN first adapts to the task via the transformation block and then learns a task-invariant representation via the task-independent embedding block.

Learning from heterogeneous feature spaces has been studied in transfer learning, or domain adaptation (Dai et al., 2008; Yang et al., 2009; Duan et al., 2012; Li et al., 2014; Yan et al., 2017; Zhou et al., 2019); see Day & Khoshgoftaar (2017) for a survey. These approaches only focus on two tasks (source and target), and require the model to learn a transformation mapping to project the source and target tasks into the same space.

Unlike meta-learning and DG methods, DEN is applicable for tasks with *heterogeneous covariates spaces*. This phenomenon is especially prevalent in tabular data tasks, where the number and definition of features could be vastly different across tasks. Iwata & Kumagai (2020) is among the first works that combine meta-learning with heterogeneous covariate spaces. Both their approach and DEN rely on pooling to handle variable-length inputs, using building blocks such as Deep Sets (Zaheer et al., 2017) and Set Transformers (Lee et al., 2019). There are several differences between their approach and DEN. Firstly, DEN uses a covariate transformation block, allowing it adapt to new tasks more efficiently. Secondly, their model is permutation invariant in covariates and thus restrictive in model expressiveness; while DEN does not have this restriction. Thirdly, we also provide theoretical insights and justification for our model architecture design.

## 3    Notations

Let $\mathbb{T}_1, \ldots, \mathbb{T}_M$ be $M$ training tasks. For each training task $\mathbb{T}$, we observe an i.i.d. sample $\mathcal{D}_\mathbb{T} = \{(\boldsymbol{x}_{\mathbb{T},i}, y_{\mathbb{T},i})\}_{i \in [n_\mathbb{T}]}$ from some joint distribution $P_\mathbb{T}$, where $[n] = \{1, \ldots, n\}$ and $\boldsymbol{x}_{\mathbb{T},i} \in \mathbb{R}^{d_\mathbb{T}}$ is the covariate vector of the $i$-th example and $y_{\mathbb{T},i} \in [L_\mathbb{T}]$ is its associated label. We denote this sample in matrix form by $(\boldsymbol{X}_\mathbb{T}, \boldsymbol{y}_\mathbb{T})$, where the $j$-th column $\boldsymbol{x}_\mathbb{T}^j \in \mathbb{R}^{n_\mathbb{T}}$ is the $j$-th covariate vector. We let $\boldsymbol{X}_{\mathbb{T},k}$ be the covariate sub-matrix corresponding to examples with label $k$ for $k \in [L_\mathbb{T}]$. When the context is clear, we drop the dependency on $\mathbb{T}$ for simplicity of the notation, e.g., we write $\boldsymbol{X}_{\mathbb{T},k}$ as $\boldsymbol{X}_k$.

Let $\mathbb{S}$ be a target task that is not contained in the training tasks. We are given a set of labeled examples $(\boldsymbol{X}_\mathbb{S}, \boldsymbol{y}_\mathbb{S})$, where the sample size $n_\mathbb{S}$ is small. We refer to it as the *support set*. The goal is to predict labels for unlabeled examples in the target task, which is called the *query set*. We denote $\mathcal{T} = \{\mathbb{T}_1, \ldots, \mathbb{T}_M, \mathbb{S}\}$.

## 4    Distribution Embedding Networks

We first describe the model architecture of DEN for binary classification in Section 4.1, and then extend DEN for multi-class classification in Section 4.2. Finally, we provide insights into the model architecture and justify its design in Section 4.3.

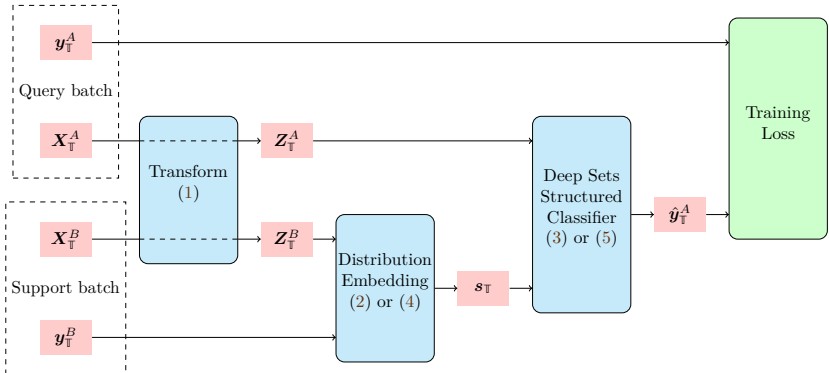

Figure 2: Block diagram of DEN for binary classification. During pre-training, for each gradient step we sample task $\mathbb{T} \in \{\mathbb{T}_1, \ldots, \mathbb{T}_M\}$ and two batches of data from the task $(\boldsymbol{X}_\mathbb{T}^A, \boldsymbol{y}_\mathbb{T}^A)$ and $(\boldsymbol{X}_\mathbb{T}^B, \boldsymbol{y}_\mathbb{T}^B)$. During fine-tuning, we treat the support set as the support batch, using which to derive distribution embedding to make predictions on the query set, treated as the query batch.

### 4.1 Model Architecture for Binary Classification

To describe the model architecture of DEN for binary classification (illustrated in Figure 2), consider data $(\boldsymbol{X}, \boldsymbol{y})$ in a given task $\mathbb{T}$. DEN can be decomposed into three major blocks: transformation, embedding and classification. We will describe these blocks in this section.

**Transforming covariates with task-dependent transformation block.** We first transform the co-variates via a transformation block, i.e.,

$$\boldsymbol{Z} = c(\boldsymbol{X}), \tag{1}$$

where $c : \mathbb{R}^d \to \mathbb{R}^d$ is applied to each row. Specifically, we use a piecewise linear function[1] (PLF) for each covariate, i.e., $c(\boldsymbol{x}) = (c^1(x^1), \ldots, c^d(x^d))$. PLFs can be optionally constrained to be monotonic, which would serve as a form of regularization during training (Gupta et al., 2016). Note that the transformation block is task-dependent—its parameters need to be re-trained for each new task. The goal is that, after applying the corresponding transformation to each task, the relatedness across tasks increases. In contrast, existing meta-learning approaches usually do not have the transformation block and thus require the raw tasks to be related. This is conceptually similar to Muandet et al. (2013), where they consider the domain generalization problem and directly learn an invariant feature transformation by minimizing the dissimilarity across domains. To the contrary, we incorporate this block into a larger architecture and learn it in an end-to-end fashion.

One may instead consider other architectures than PLFs for the transformation block. We choose PLFs since they can implement compact one-dimensional non-linearities and can thus be fine-tuned with a small support set. Moreover, they are universal approximators: with enough keypoints, they can approximate any one-dimensional bounded continuous functions. In Section 5.3 we show that this PLF transformation is key to ensure good performance when training and target tasks have heterogeneous covariate spaces. PLFs cannot model interactions between covariates, which is a sacrifice we may have to make in light of the small support set. We study in Section 5.3 the trade-offs between the flexibility of PLF and the size of the support set.

**Summarizing task statistics with task-independent distribution embedding block.** The second block in DEN is to learn a vector that summarizes the task distribution. This is similar to Garnelo et al. (2018). Naïvely, one could learn a non-linear transformation $\phi$ which embeds $(\boldsymbol{z}, y)$ into a vector of smaller dimension. However, this would not work since the dimension of $\boldsymbol{z}$ can vary across tasks. In contrast, we embed the distribution $P(\boldsymbol{z}, y)$ of a given task into a vector using the transformed features $\boldsymbol{Z}$ in the following

---

[1]See Appendix A for the precise definition.

way. For all $a, b \in [d]$, we derive a distribution embedding of $P(z^a, z^b, y)$ by

$$s^{a,b} = \left( \overline{h(\boldsymbol{Z}_1^{a,b})}, \overline{h(\boldsymbol{Z}_2^{a,b})}, \overline{\boldsymbol{y}} \right), \tag{2}$$

where $h$ is a vector-valued *trainable* function, and the average is taken with respect to the training batch during training and support set during inference, i.e.,

$$\overline{h(\boldsymbol{Z}_k^{a,b})} = \frac{\sum_{i=1}^n h(z_i^a, z_i^b) \mathbf{1}\{y_i = k\}}{\sum_{i=1}^n \mathbf{1}\{y_i = k\}},$$

and $\overline{\boldsymbol{y}} = \frac{1}{n} \sum_{i=1}^n y_i$. Note that $n$ is about 50 in our setting, so the empirical average is close to the population counterpart. Intuitively, we decompose a variable-length feature vector $\boldsymbol{z}$ into smaller pieces of fixed length 2, and use the $h$ function to learn a pairwise embedding $s^{a,b}$ for each pair of the 2 features $a$ and $b$. This pairwise decomposition allows us to handle variable-length covariates. Note that the same function $h$ is shared across all tasks, and it can be chosen as a few fully connected layers. The distribution embedding of $P(\boldsymbol{z}, y)$ is thus the set of embeddings of all pairs $\boldsymbol{s} = \left( s^{a,b} \right)_{a,b \in [d]}$.

*Remark* 4.1. The length 2 here is arbitrary. We can use pieces of length $r$ for any $r \geq 1$, i.e., obtain an embedding $s^{t_1, \ldots, t_r}$ of $P(\boldsymbol{z}^{t_1, \ldots, t_r}, y)$ for all $t_1, \ldots, t_r \in [d]$. The larger $r$ is the more expressive the model will be. We refer to $r$ as the *dependency order*, and experiment with different values in Section 5.3.

**Prediction with task-independent classification block.** Given a query $\boldsymbol{x}$, we first transform it via $\boldsymbol{z} = c(\boldsymbol{x})$, and decompose the transformed features into sets of feature pairs. We then obtain the distribution embedding vector $s^{a,b}$ of each pair in (2). Finally, we obtain the predicted logits using a Deep Sets architecture (Zaheer et al., 2017):

$$q = \Phi(\boldsymbol{z}, \boldsymbol{s}) = \psi \left( \sum_{a, b \in [d]} \varphi \left( \left[ \boldsymbol{z}^{a,b}, s^{a,b} \right] \right) \right), \tag{3}$$

where $\varphi$ is a vector-valued trainable function and $\psi$ is a real-valued trainable function. Both $\varphi$ and $\psi$ are shared across tasks and can be chosen as fully connected layers. The Deep Sets architecture, which aggregates all possible pairs of covariates, is proven to be a universal approximator of set-input functions (Zaheer et al., 2017). We note that one may use other set input architectures to construct the classification block, e.g., Set Transformer (Lee et al., 2019).

## 4.2 Model Architecture for Multiclass Classification

For multiclass classification tasks, we modify the distribution embedding and classification blocks. Specifically, we modify the distribution embedding in (2) as

$$s^{a,b} = \frac{1}{n} \sum_{i=1}^n h \left( \left[ z_i^a, z_i^b, \boldsymbol{v}(y_i) \right] \right), \tag{4}$$

where $\boldsymbol{v} : \mathbb{N} \to \mathbb{R}^m$ is a vector-valued, trainable function that is shared across tasks—$\boldsymbol{v}(k)$ is hence a vector encoding of class $k$—and $h$ is a vector-valued trainable function with input dimension $m + 2$.

For the classification block, we modify the idea of Matching Net (Vinyals et al., 2016), which is also similar to the modification adopted by Iwata & Kumagai (2020). This modification is suitable for tasks with different numbers of label classes. Let $\tilde{\Phi}$ be $\Phi$ in (3) without the last layer, i.e., $\tilde{\Phi}(\boldsymbol{z}, \boldsymbol{s}) = \sum_{a,b \in [d]} \varphi([\boldsymbol{z}^{a,b}, s^{a,b}])$ is the penultimate layer embedding of the query set example. We then obtain its class scores by

$$q_k = \frac{\sum_{i=1}^n \tilde{\Phi}(\boldsymbol{z}, \boldsymbol{s})^\top \tilde{\Phi}(\boldsymbol{z}_i, \boldsymbol{s}) \mathbf{1}\{y_i = k\}}{\sum_{i=1}^n \mathbf{1}\{y_i = k\}}. \tag{5}$$

Note that $i$ in the above equation is the index of support set examples ($n$ in total). The score $q_k$ can thus be interpreted as the average dot-product of the penultimate layer embedding of the given query example with the penultimate embedding of the support set examples of class $k$. To obtain class probability, we apply a softmax on $q_k$'s.

### 4.3 Rationale for the Architectural Design

For further insights into the model architecture, consider the optimal Bayes classifier:

$$P(y = k \mid \boldsymbol{x}) = \frac{P(\boldsymbol{x} \mid y = k)P(y = k)}{\sum_{l=1}^{L} P(\boldsymbol{x} \mid y = l)P(y = l)}. \tag{6}$$

For a given task, if the conditional probability $P(\boldsymbol{x} \mid y = k)$ belongs to the same family of distributions for all $k \in [L]$, i.e., $\boldsymbol{x} \mid y = k \sim \phi(\boldsymbol{x}; \boldsymbol{\theta}_k)$, then the Bayes classifier can be constructed by estimating the parameters $\boldsymbol{\theta}_k$ and $P(y = k)$, and approximating the density $\phi$ as

$$P(y = k \mid \boldsymbol{x}) = \frac{\phi(\boldsymbol{x}; \hat{\boldsymbol{\theta}}_k)\hat{P}(y = k)}{\sum_{l=1}^{L} \phi(\boldsymbol{x}; \hat{\boldsymbol{\theta}}_l)\hat{P}(y = l)}. \tag{7}$$

If, additionally, $P(\boldsymbol{x} \mid y = k)$ belongs to the same family of distributions for all $k \in [L]$ and also all tasks, then the Bayes classifers for all tasks should have the same functional form; only the parameters $\boldsymbol{\theta}_k$ and $P(y = k)$ differ by task. In this case, we can simply pool the data from all tasks together to estimate $\boldsymbol{\theta}_k$, $P(y = k)$ for each task, and the task-independent function $\phi$.

However, the distribution family $\phi(\boldsymbol{x}; \boldsymbol{\theta}_k)$ may vary greatly across tasks. The task-dependent transformation block allows the transformed covariates to be in the same distribution family approximately. Then, we utilize the distribution embedding block to estimate the parameters $\boldsymbol{\theta}_k$ and $P(y = k)$. If all transformed covariates belong to the same distribution family, then the function form to estimate the parameters $\boldsymbol{\theta}_k$ and $P(y = k)$ should be identical for all tasks, which justifies our use of a *task-independent* distribution embedding block. Finally, we use the task-independent classification block to approximate the task-independent function $\phi$ and obtain a score for each label. The effectiveness of DEN depends crucially on the ability of the transformation block to align distribution families of covariates across tasks. We study in Section 5.3 the performance of DEN in relation to the flexibility of PLFs.

Finally, since the covariate dimension can vary across tasks, we decompose the covariate vector into sub-vectors of fixed length $r$ and apply a Deep Sets architecture to these sub-vectors. In fact, Proposition 4.3 shows that if we consider the following family of densities, then the Bayes classifier must be of the form (3).

**Definition 4.2.** Let $\{f(\cdot; \boldsymbol{\theta}) : \mathbb{R}^r \to \mathbb{R}\}$ be a parametric family of functions (not necessarily densities). For any integer $d \geq r$, we say a function $g$ on $\mathbb{R}^d$ admits an $f$-*expansion* if it factorizes as $g(\boldsymbol{z}) = \prod_{t_{1:r} \in [d]^r} f(\boldsymbol{z}^{t_{1:r}}; \boldsymbol{\theta}^{t_{1:r}})$, where $\{\boldsymbol{\theta}^{t_{1:r}} \in \mathbb{R}^\tau\}$ is a set of parameters.

For instance, if $\boldsymbol{z}|y = 1 \sim \mathcal{N}_d(\boldsymbol{\mu}, \sigma^2 I_d)$, then the conditional density $p(\boldsymbol{z}|y = 1)$ is proportional to

$$\prod_{a=1}^{d} \frac{1}{\sigma} \exp\left(-\frac{(z^a - \mu^a)^2}{2\sigma^2}\right),$$

which admits an $f$-expansion with $r = 1$ and $\boldsymbol{\theta}^a = (\mu^a, \sigma)$.

**Proposition 4.3.** *Let $(\boldsymbol{z}, y)$ be a random vector in $\mathbb{R}^d \times [L]$ following some distribution $P$. Assume that the conditional density $p(\boldsymbol{z}|y = k)$ admits an $f$-expansion for some parametric family of functions $\{f(\cdot; \boldsymbol{\theta})\}$ on $\mathbb{R}^r$ with parameters $\{\boldsymbol{\theta}_k^{t_{1:r}}\}$. Then there exist functions $\psi$ and $\varphi$ such that*

$$P(y = k \mid \boldsymbol{z}) \propto \psi\left(\sum_{t_{1:r} \in [d]^r} \varphi(\boldsymbol{z}^{t_{1:r}}, \boldsymbol{\theta}_k^{t_{1:r}}, \pi_k)\right), \tag{8}$$

*where $\boldsymbol{z}^{t_{1:r}} = (z^{t_1}, \dots, z^{t_r})$, $\pi_k = P(y = k)$, and $\psi$ and $\varphi$ only depend on $f$.*

The proof is included in Appendix A. Note that our model (3) has exactly the same structure as the optimal Bayes classifier in (8) with $r = 2$ and $\boldsymbol{s}$ representing the parameters $\{\boldsymbol{\theta}_k^{t_{1:r}}\}$ and marginal $\{\pi_k\}$. Proposition 4.3 shows that, under appropriate conditions, our model class is expressive enough to include the optimal Bayes classifier. This justifies our choice of the distribution embedding (2) and the Deep Sets structure (3). DEN will consequently performed well when learning across tasks whose conditional distributions of the PLF-transformed feature admits the same $f$-expansion. Heuristically, this means DEN is ideally applied to meta-learning settings in which features across tasks can be transformed to have a similar structure.

Table 1: Overview of training and evaluation.

| Step | Input data | Transform | Embedding | Classification |
|------|-----------|-----------|-----------|----------------|
| Pre-training | Heterogeneous tasks $\{(\boldsymbol{x}_{\mathbb{T},i}, y_{\mathbb{T},i})\}_{\mathbb{T},i}$ | Trained | Trained | Trained |
| Fine-tuning | Support set $\{(\boldsymbol{x}_{\mathbb{S},i}, y_{\mathbb{S},i})\}_i$ | Trained $c_{\mathbb{S}}$ | Fixed | Fixed |
| Evaluation | Query $\boldsymbol{x}_{\mathbb{S},q}$ and support set $\{(\boldsymbol{x}_{\mathbb{S},i}, y_{\mathbb{S},i})\}_i$ | Fixed $c_{\mathbb{S}}$ | Fixed | Fixed |

### 4.4 Training and Inference

Figure 2 shows a high level summary of our three-block model architecture. The overall training and evaluation procedure is summarized in Table 1. Note that all sets of $r$ inputs use the same $h$ and $\varphi$ functions through parameter sharing, which reduces the size of the model. For DEN applied on tasks with $d$ features, the feature transformation block has $\mathcal{O}(d)$ parameters. An embedding block with embedding order $r$, $L$ layers and $H$ hidden nodes per layer has $rH + H^2(L-1)$ parameters. A classification block with $L$ layers and $H$ hidden nodes per layer for both the $\varphi$ and $\psi$ functions has $(r+2)H + 2H^2L$ parameters. This DEN model is roughly of the same size as a $3L$-layer deep $H$-node wide neural network. Empirically, we found that the computational complexity and resource requirement of DEN are similar to those of Vinyals et al. (2016) and Iwata & Kumagai (2020) given similar model sizes.

During pre-training, in each gradient step, we randomly sample a task $\mathbb{T} \in \{\mathbb{T}_t\}_{t=1}^M$ and two batches $A$ and $B$ from $(\boldsymbol{X}_{\mathbb{T}}, \boldsymbol{y}_{\mathbb{T}})$. These two batches are first transformed using PLFs in (1). We then use (2) or (4) to obtain a distribution embedding $\boldsymbol{s}_{\mathbb{T}}$, taking the average with respect to the examples in the support batch. Next, we use the distribution embedding to make predictions on the query batch using (3) or (5). Note that during training, $\boldsymbol{s}_{\mathbb{T}}$ is identical across examples within the same batch, but it could vary (even within the same task) across batches.

If PLFs can approximately transform the covariates so that they admit the same $f$-expansion, then the rest of the network is task-independent. Thus, after pre-training on $\{\mathbb{T}_t\}_{t=1}^M$, for each new task $\mathbb{S}$, we can fine-tune the transformation block (1), while keeping the weights of other blocks fixed. Because PLFs only have a small number of parameters, they can be trained on a small support set from the task $\mathbb{S}$.

During inference, we first use the learned PLFs in (1) to transform covariates in both the support set and the query set. We then utilize the learned distribution embedding block to obtain $\boldsymbol{s}_{\mathbb{S}}$, where the average in (2) and (4) is taken over the whole support set. Finally, the embedding $\boldsymbol{s}_{\mathbb{S}}$ and PLF-transformed query set covariates are used to classify query set examples using (3) or (5).

## 5 Numerical Studies

In Section 5.1, we use OpenML tabular datasets to demonstrate the performance of DEN on real-world tasks. DEN achieves the best performance among a number of baseline methods. We then introduce in Section 5.2 an approach to simulate binary classification tasks, which allows us to generate a huge amount of pre-training examples without the need of collecting real tasks. Surprisingly, DEN (and several other methods) trained on simulated data can sometimes outperform those trained on real data. In Section 5.3, we examine the performance of DEN in relation to different architecture and hyper-parameter choices. The findings are: (a) PLF and fine-tuning are crucial when the training and test tasks are unrelated (e.g., when training tasks are simulated), whereas their effect is insignificant when the tasks are similar, and (b) the performance of DEN is relatively stable for small values of the dependency order $r$ in Remark 4.1.

Baseline methods for comparison with DEN include Matching Net (Vinyals et al., 2016), Proto Net (Snell et al., 2017), TADAM (Oreshkin et al., 2018), PMN (Wang et al., 2018), Relation Net (Sung et al., 2018), CNP (Garnelo et al., 2018), MAML (Finn et al., 2017), BMAML (Finn et al., 2018), T-Net (Lee & Choi, 2018) and Iwata & Kumagai (2020). Hyperparameters of all methods are chosen based on cross-validation on training tasks. Hyperparameters, model structures and implementation details are summarized in Appendix B. Note that we set the dependency order $r = 2$ if it is not stated explicitly. For MAML, BMAML and T-net,

Table 2: Average test AUC (standard error) and the percent of times that each method achieves the best performance on 20 OpenML datasets × 20 repeats.

| Method | Average Test AUC (%) | % Best |
|---|---|---|
| Matching Net | 50.11 (0.04) | 0.00% |
| Proto Net | **71.11** (0.72) | 27.50% |
| PMN | 56.11 (0.65) | 0.75% |
| Relation Net | 51.65 (0.28) | 0.25% |
| CNP | 58.01 (0.69) | 7.00% |
| Iwata & Kumagai (2020) | **70.35** (0.70) | 26.75% |
| MAML | 60.64 (0.82) | 7.00% |
| T-Net | 52.22 (0.41) | 0.5% |
| DEN | **70.12** (0.83) | **30.25%** |

we fine-tune the last layer of the base model for 5 epochs on the support set. For the rest of the methods, we train them with episodic training[2] (Vinyals et al., 2016). For methods that do not readily handle variable length inputs, we randomly repeat features to make all tasks have the same length of inputs.

### 5.1 Results on OpenML Classification Tasks

#### 5.1.1 Binary classification

We compare DEN with baseline methods on 20 OpenML binary classification datasets (Vanschoren et al., 2013) following the setup in Iwata & Kumagai (2020) (see a list of datasets in Appendix C). These datasets have examples ranging from 200 to 1,000,000 and features ranging from 2 to 25.

We pre-train DEN and baseline methods on the OpenML datasets in the leave-one-dataset-out fashion. That is, for each of the 20 OpenML datasets chosen as a target task, we pretrain the models on the remaining 19 datasets. For the target task, we randomly select 50 examples to form the support set, and use the rest of the dataset as the query set. We repeat the whole procedure 20 times, and report the average AUC and the percentage that each method achieves the best performance in Table 2 based on 20 test sets × 20 repeats. DEN has average AUC comparable with the best methods, and the highest frequency that it achieves the best AUC. As a comparison, directly training task-specific linear models on the support set of each task gives an average test AUC of 57.15%. Directly training an 1 hidden layer, 8 hidden nodes neural network on the support set gives an average test AUC of 54.58%. With 2 hidden layers, AUC drops to 53.00% and with 3 hidden layers, 49.74%. DEN significantly outperforms those methods. These results demonstrate the effect of over-fitting for classical methods on small data, and the benefit of DEN over classical methods on those small-data problems. In Section 5.2, we will further describe an approach to generate binary classification training tasks through controlled simulation. DEN trained on simulated tasks, surprisingly, outperforms DEN trained on real tasks, and, in fact, achieves the best performance among all competing methods.

#### 5.1.2 Multiclass classification

In addition to binary classification tasks, we also compare DEN with baseline methods on 8 OpenML multi-class classification datasets. These datasets have examples ranging from 400 to 1,000,000, features ranging from 5 to 23, and number of classes ranging from 3 to 7. We train DEN and baseline methods on the OpenML datasets in the same leave-one-dataset-out fashion as in the binary classification. Results are summarized in Table 3. DEN achieved the best performance, followed by Matching Net. We also compare against directly training a neural network (NN) on the support set, which achieved decent accuracy. But DEN remains the best methods in majority (61.6%) of cases.

---

[2]Code is available at https://github.com/google-research/google-research/distribution_embedding_networks.

Table 3: Average test accuracy (standard error) and the percent of times that each method achieves the best performance on 8 OpenML datasets $\times$ 20 repeats.

| Method | Average Test Accuracy (%) | % Best |
|---|---|---|
| Direct NN | **45.68** (1.56) | 13.6% |
| Matching Net | **46.36** (1.55) | 13.8% |
| Proto Net | 33.68 (1.80) | 4.9% |
| PMN | 24.77 (0.63) | 0.0% |
| Relation Net | 33.49 (1.82) | 4.9% |
| CNP | 18.77 (1.53) | 1.1% |
| Iwata & Kumagai (2020) | 24.93 (0.62) | 0.0% |
| DEN | **48.60** (1.51) | **61.6%** |

## 5.2 Generate Training Tasks through Controlled Simulation

In this section, we describe an approach to generate binary classification pre-training tasks based on model aggregation. Comparing to pre-training meta-learning models on related real-world datasets, which could be expensive to collect, this synthetic approach can easily give us a huge amount of pre-training examples. We will show that meta-learning methods trained with simulated data can, surprisingly, sometimes outperform those trained with real data.

Specifically, we first take seven image classification datasets: CIFAR-10, CIFAR-100 (Krizhevsky, 2009), MNIST (LeCun et al., 2010), Fashion MNIST (Xiao et al., 2017), EMNIST (Cohen et al., 2017), Kuzushiji MNIST (KMNIST; Clanuwat et al., 2018) and SVHN (Netzer et al., 2011). On each dataset, we pick nine equally spaced cutoffs and binarize the labels based on whether the class id is below the cutoff. This gives nine binary classification tasks for each dataset with positive label proportion in $\{0.1, 0.2, \ldots, 0.9\}$.

To generate covariates for each task, we build 50 convolution image classifiers of various model complexities (details in Appendix B) on each of the 63 tasks to predict the binary label. We take classification scores on the test set as covariates. With these covariates and associated labels, we construct $7 \times 9 = 63$ binary classification tasks $\{\mathbb{T}_1, \ldots, \mathbb{T}_{63}\}$. Essentially, they are model aggregation tasks since we are combining 50 classifiers to make a prediction. Note that the accuracy of those image classifiers ranges from below 0.6 to over 0.99, giving rise to covariates ranging widely in their signal-to-noise ratios.

Finally, to augment training data, we apply covariate sampling during pre-training. In each pre-training step, we first randomly sample an integer $C \in \{1, \ldots, 50\}$ and a task from the 63 aggregation tasks $\{\mathbb{T}_1, \ldots, \mathbb{T}_{63}\}$ described above. Then, among the 50 convolution image classifiers we built, we randomly pick $C$ of them and use their classification scores as covariates to construct a *sub-task*. Finally, DEN takes labeled examples from this sub-task and uses them for training in this step. We shall emphasize that although training tasks are built based on image classification datasets, we do *not* use raw pixel values as covariates in pre-training.

**OpenML binary classification with simulated training tasks.** To illustrate the effectiveness of our data simulation approach, we use the models pre-trained on these data and evaluate them on the same 20 OpenML binary classification tasks in Section 5.1 after fine-tuning. Interestingly, as shown in Table 4, for 5 out of the 9 methods considered, pre-training on the simulated data gives us statistically significantly better test AUC. This suggests that the proposed approach to generate training tasks is not only convenient but also effective. Moreover, DEN pre-trained on the simulated data outperforms all methods (either pre-trained on the simulated data or OpenML data) significantly.

## 5.3 Effect of Dependence Order, PLF, and Fine-tuning

We continue our examination of DEN with pre-training on simulated data. In addition to comparing DEN with competing methods, we also study the performance of DEN in relation to its dependence order $r$, and

Table 4: Average test AUC (standard error) and the percent of times that each method achieves the best performance on 20 OpenML datasets × 20 repeats. The models are pre-trained on simulated data. We also report the improvement in average test AUC compared to pre-training on OpenML data.

| Method | Average Test AUC (%) | % Best | Improv. in Test AUC (%) |
|---|---|---|---|
| Matching Net | 53.88 (0.46) | 0.00% | 3.77 (0.46) |
| Proto Net | 71.12 (0.65) | 28.00% | 0.01 (0.97) |
| PMN | 59.04 (0.68) | 1.25% | 2.93 (0.94) |
| Relation Net | 57.97 (0.59) | 0.00% | 6.32 (0.65) |
| CNP | 60.95 (0.69) | 2.50% | 2.94 (0.98) |
| Iwata & Kumagai (2020) | 66.01 (0.74) | 11.50% | -4.34 (1.02) |
| MAML | 61.16 (0.66) | 2.75% | 0.52 (1.05) |
| T-Net | 53.35 (0.46) | 0.25% | 1.13 (0.62) |
| DEN | **74.13** (0.68) | **53.75%** | 4.01 (1.07) |

Table 5: Average test AUC (standard error) on Nomao and Puzzles data.

| Method | Nomao | Puzzles |
|---|---|---|
| Matching Net | 73.18 (2.23) | 62.65 (1.45) |
| Proto Net | 80.56 (0.56) | 73.77 (0.37) |
| TADAM | 82.42 (0.35) | 74.86 (0.25) |
| PMN | 77.00 (3.13) | 57.69 (1.53) |
| Relation Net | 52.32 (1.61) | 63.73 (1.79) |
| CNP | 91.40 (0.36) | 53.20 (0.12) |
| Iwata & Kumagai (2020) | 66.85 (3.23) | 61.32 (0.52) |
| MAML | 78.92 (2.22) | 54.92 (0.54) |
| BMAML | 47.20 (3.40) | 53.49 (1.88) |
| T-Net | 61.42 (3.44) | 54.55 (1.78) |
| DEN w/o PLF w/o FT | 59.01 (1.04) | 68.87 (0.38) |
| DEN w/o FT | 94.42 (0.16) | 70.74 (0.62) |
| DEN | **95.21** (0.10) | **78.11** (0.53) |

the use of PLF and fine-tuning. For DEN, we explore three options: 1) fine-tune the PLF layer for 10 epochs, 2) take the PLF layer directly from the last pre-training epoch without fine-tuning, or 3) do not include a PLF layer in DEN and hence no fine-tuning at all.

### 5.3.1 Tasks with Heterogeneous Covariate Spaces

We study and demonstrate the importance of PLF when the training and test tasks are heterogeneous. Specifically, we use all 63 simulated tasks described in Section 5.2 for pre-training, and test the performance on two real datasets: Nomao and Puzzles. We give a short description of each dataset in Section C.

For each dataset, we repeat the whole procedure 20 times and report the average AUC and standard error in Table 5. It is clear that DEN outperforms other baseline methods significantly. More importantly, the results also show that fine-tuning and PLF greatly improve the performance of DEN. Since DEN is pre-trained on simulated tasks which are completely unrelated to the target task, this improvement demonstrates the importance of fine-tuning and PLF when the training and target tasks are heterogeneous.

We also examine the effect of the size of the support set on the flexibility of the covariate transformation block. Figure 3 shows that with enough support set examples (e.g., Puzzles task), having more PLF keypoints

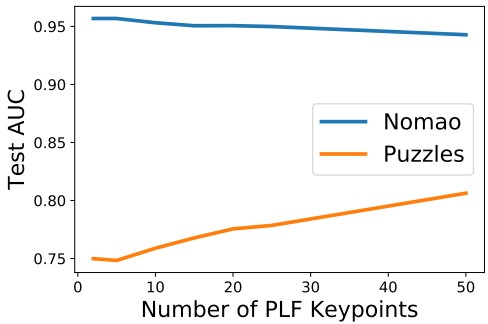

Figure 3: Average test AUC versus the number of PLF keypoints on Nomao and Puzzles data.

Table 6: Average test AUC (standard error) of DEN on Nomao data with different dependency order.

| $r = 1$ | $r = 2$ | $r = 3$ | $r = 4$ |
|---|---|---|---|
| 94.38 (0.46) | **95.21** (0.10) | 93.80 (0.25) | 92.61 (0.55) |
| $r = 5$ | $r = 6$ | $r = 7$ | $r = 8$ |
| 93.16 (0.45) | 91.22 (0.74) | 88.95 (2.02) | 89.15 (1.34) |

benefits the test performance due to the improved ability of task-adaptation. However, if the support set is small (e.g., Nomao task), having a flexible covariate transformation block could even be marginally harmful.

Finally, we conduct an ablation study to examine the effect of dependency order in Remark 4.1 on the test AUC on the Nomao dataset. In particular, we examine DEN with $r \in [8]$. The results in Table 6 show that when $r \leq 5$, the test AUC is relatively stable, with the best performance achieved at $r = 2$; whereas the test AUC is much worse and unstable for larger $r$.

### 5.3.2 Tasks with Homogeneous Covariate Spaces

We study the performance of DEN in the case when the training and target tasks are homogeneous. To ensure the task homogeneity, we use the model aggregation tasks described in Section 5.2 for both training and evaluation. Specifically, we use the $5 \times 9 = 45$ tasks described in Section 5.2 derived from CIFAR-10, CIFAR-100, MNIST, Fashion MNIST and EMNIST to train DEN and other meta-learning methods. We then pick four test tasks from SVHN and KMNIST of different difficulties, where the average AUCs over 50 classifiers are 68.28%, 78.11%, 91.51%, and 87.58%, respectively. Note that the training and test tasks are totally separated, but they likely follow similar distributions since the covariates for training and tests are all classifier scores.

For each test task, we randomly select 100 sets of $C$ classifiers among the 50 candidate classifiers, resulting in 100 aggregation sub-tasks. For each aggregation sub-task, we form a support set with 50 labeled examples and a disjoint query set with 8000 examples. We repeat the entire training and fine-tuning process for 5 times, and report the average AUC and its standard error.

Table 7 shows the result of aggregating an ensemble of $C = 25$ classifiers. Table 8 shows the result where the number of classifiers $C$ to be aggregated is sampled uniformly from $[13, 25]$ and could vary across sub-tasks (100 aggregation sub-tasks $\times$ 5 repeats). To allow baseline methods to take varying number of covariates, we randomly duplicate some of the $C$ classifiers so that all inputs have 25 covariates.

We observe that DEN significantly outperforms other methods in all tasks, and that DEN without PLF and/or without fine-tuning is statistically no worse than DEN with fine-tuning on the PLF layer. This suggests that fine-tuning the PLF layer is not necessary when the data distribution is similar among tasks.

Table 7: Average test AUC (standard error) when aggregating 25 classifiers.

| Method | Test AUC (%) | | | |
| --- | --- | --- | --- | --- |
| | Task 1 | Task 2 | Task 3 | Task 4 |
| Matching Net | 70.01 (0.40) | 82.13 (0.05) | 95.29 (0.06) | 93.82 (0.08) |
| Proto Net | 90.95 (0.05) | 89.84 (0.03) | 98.07 (0.01) | 97.40 (0.02) |
| TADAM | 90.98 (0.05) | 89.90 (0.03) | 98.14 (0.01) | 97.57 (0.02) |
| PMN | 86.78 (0.10) | 88.69 (0.03) | 97.46 (0.02) | 96.22 (0.05) |
| Relation Net | 85.39 (0.15) | 88.70 (0.02) | 97.25 (0.02) | 95.55 (0.08) |
| CNP | 86.53 (0.09) | 88.80 (0.02) | 97.50 (0.02) | 96.22 (0.05) |
| Iwata & Kumagai (2020) | 89.33 (0.05) | 89.17 (0.02) | 97.93 (0.01) | 97.73 (0.02) |
| MAML | 86.10 (0.11) | 88.78 (0.03) | 97.48 (0.02) | 96.13 (0.06) |
| BMAML | 71.38 (0.84) | 85.96 (0.21) | 97.04 (0.08) | 95.39 (0.18) |
| T-Net | 86.23 (0.10) | 88.76 (0.03) | 97.47 (0.02) | 96.11 (0.06) |
| DEN w/o PLF w/o Fine-Tuning | 91.53 (0.03) | **90.18** (0.02) | 98.03 (0.01) | 98.37 (0.01) |
| DEN w/o Fine-Tuining | **91.76** (0.03) | **90.20** (0.02) | **98.18** (0.01) | **98.41** (0.01) |
| DEN | **91.80** (0.03) | 89.77 (0.02) | 97.38 (0.01) | 97.23 (0.01) |

Table 8: Average test AUC (standard error) when aggregating variable number of classifiers.

| Method | Test AUC (%) | | | |
| --- | --- | --- | --- | --- |
| | Task 1 | Task 2 | Task 3 | Task 4 |
| Matching Net | 75.16 (0.37) | 81.12 (0.08) | 95.86 (0.04) | 93.61 (0.08) |
| Proto Net | 90.02 (0.11) | 89.65 (0.04) | 97.94 (0.02) | 97.57 (0.02) |
| TADAM | 90.20 (0.10) | 89.74 (0.04) | 98.04 (0.01) | 97.84 (0.01) |
| PMN | 85.68 (0.24) | 88.59 (0.04) | 97.30 (0.03) | 95.83 (0.08) |
| Relation Net | 80.85 (0.65) | 88.53 (0.04) | 97.11 (0.04) | 95.03 (0.12) |
| CNP | 84.97 (0.28) | 88.71 (0.04) | 97.31 (0.04) | 95.96 (0.07) |
| Iwata & Kumagai (2020) | 89.26 (0.16) | 89.05 (0.03) | 97.82 (0.02) | 97.75 (0.02) |
| MAML | 85.39 (0.23) | 88.53 (0.04) | 97.32 (0.04) | 95.86 (0.08) |
| BMAML | 59.57 (1.07) | 85.40 (0.24) | 96.73 (0.10) | 94.53 (0.23) |
| T-Net | 85.51 (0.22) | 88.55 (0.04) | 97.35 (0.03) | 95.94 (0.07) |
| DEN w/o PLF w/o Fine-Tuning | 91.06 (0.09) | **89.92** (0.03) | **98.09** (0.01) | **98.24** (0.01) |
| DEN w/o Fine-Tuning | **91.17** (0.09) | **89.95** (0.03) | 97.95 (0.01) | 98.10 (0.01) |
| DEN | **91.29** (0.03) | 89.83 (0.02) | 97.60 (0.03) | 97.47 (0.01) |

## 6 Conclusion

In this work, we introduce a novel meta-learning method that can be applied to settings where both the distribution and number of covariates vary across tasks. This allows us to train the model on a wider range of training tasks and then adapt it to a variety of target tasks. Most other meta-learning techniques do not readily handle such settings. In numerical studies, we demonstrate that the proposed method outperforms a number of meta-learning baselines.

The proposed model consists of three flexible building blocks. Each block can be replaced by more advanced structures to further improve its performance. DEN can also be combined with optimization based meta-learning methods, e.g., MAML. A limitation of DEN is that it requires calculating the embedding for $d^r$ different combinations of covariates, which is infeasible for high-dimensional tasks. A potential solution is to use a random subset of these combinations. We leave the exploration of these options for future work.

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
