# OpenReview forum: "Distribution Embedding Networks for Generalization from a Diverse Set of Classification Tasks"
_TMLR — Accepted by TMLR_

### Review · Reviewer_gQmZ · 2022-07-22

**Summary Of Contributions:**

This paper tracks an important problem in meta-learning: the tasks are not necessarily related. By designing the task specific representation, this paper jointly adopted the task specific representation and nature representation Z for the prediction. The empirical results on different UCI datasets and simple image datasets demonstrates the effectiveness of the proposed method.


**Requested Changes:**

Indeed I really like the problem setup and the motivation, unfortunately the proposed approach seems not clearly addressing the problem. In fact, the proposed analysis seems ad-hoc and many components lack clear support or justifications. Based on these, I would suggest

- A clear explanation and justification of each component, particularly why designing a meta-representation learning approach? If it is necessary, how to fully justify the learned representation is correct ? How to justify the encoding of the task specific feature $S$ exactly to capture the correct information?   The whole proposed approach lack clear **justifications**.

- A revisit on the evaluated datasets. In general, the motivation of deep-meta learning is to learn a representation. In this paper, except accuracy, there is no other proper information/analysis about the representation learning. Why not a simple sparse-coding based approach on tabular datasets?

- The derived theory could not support the proposed framework in few-shot settings. Authors are strongly suggested to make revisions on the theory.


**Strengths And Weaknesses:**

### Strong points
1. This paper considered an important problem in meta representation learning: the tasks are not necessarily related. Then enforcing a meta-prior is not proper.
2.  The proposed method is empirically validated by different open source datasets.

### Weak points

1. The proposed approach is merely a collection of different other approaches. Although TMLR does not emphasize the novelty, this paper is still strongly suggested to illustrate the motivation of choosing these components.
2. Another main concern is why considering meta-representation learning in the context of tabular or simple image dataset. These datasets (particularly tabular dataset) could be solved by the “non-deep” regime such as kernel method or sparse coding. Based on this argument, the experiment or the proposed approach seems not convincing.
3. Concerns in the proposed analysis.

### Detailed comments

1. A major concern is the motivation of the proposed approach. Please note the motivation of the problem itself is clear and well-appreciated.  In the context of uci dataset and simple image dataset, why do we need to learn a representation through deep neural networks?  If the data is UCI or simple image, we could learn the embedding though kernel or spare coding. The corresponding method could be significantly simplified. This reviewer feels rather confused about the motivation of the proposed approach or evaluated empirical results.

2. The key to encode the task specific feature is equation (2-4), while this reviewer has multiple concerns:
- In Eq(2), the task is represented by the label specific features and label average. While this seems a bit heuristic since the different labels across the tasks do not necessarily have the same label space. E.g, for two different binary classification, the same label average is not meaningful.
- In Eq(3,4), the combination of learned embedding $Z$ and task specific embedding $S$ are concatenated, what is the support of combining information in this manner? Why not tensor product?
- Besides, why introduce a,b in the embedding S, what is the specific benefit? This makes both high memory and computational complexity. It seems this paper did NOT report the comparison of  time and memory complexity.

3. In Sec 4.3, the analysis is a bit ad-hoc, in Equation (6-7), the P(y) and P(x|y=k) are ground truth distribution. Unfortunately, in the context of this paper (few-shot settings), adopting the empirical term as an approximation of the ground truth distribution is incorrect. In fact, Sec 4.3 could not provide a clear explanation of the proposed approach. It seems an ad-hoc analysis without considering the few-shot or meta learning problems.


4. In the empirical comparison, since the data is not complex or extremely high-dimensional, comparing with modern deep network based approaches seems unfair.

---

> ### Author Response · Authors · 2022-08-27
> **One Clarifying Question**
>
> Dear Reviewer,
>
> Thanks for your careful reviews. We will post a complete and detailed response to your questions and suggestions in a few days.
>
> In the meantime, to ensure a satisfactory response, we have the following clarifying question:
>
> You asked us why we didn't use a simple sparse-coding based approach on tabular datasets. We are not sure what you referred as the "sparse-coding based approach". Could you give us an example?

---

> > ### Comment · Reviewer_gQmZ · 2022-08-28
> > **Quick follow-up**
> >
> > Thanks for your question. I mean when learning a representation, it is not necessarily to use neural-network for the tabular dataset. For instance sparse dictionary learning (such as https://en.wikipedia.org/wiki/Sparse_dictionary_learning, or Sparse Modeling for Image and Vision Processing. Foundations and Trends in Computer Graphics and Vision) could also achieve good results in the context of tabular data or simple image dataset.

---

> ### Author Response · Authors · 2022-09-01
> **Response to your general questions**
>
> Thank you for your careful review of our paper. Please see our response to your questions below, and let us know if you have further questions. We will upload a revision of this paper in the next few days.
>
> **Re: learning embedding through deep neural net rather than sparse coding or kernel methods**
>
> Sparse coding aims to embed each input as a linear combination of some sparse base representations, where the learned linear combination weight is called the dictionary. The dictionary matrix is of dimension $D\times K$, where $D$ is the input dimension and $K$ is the embedding dimension. Since the input dimension $D$ may vary across tasks, the dictionary matrix needs to be task-dependent. Because of this, there will be little “knowledge sharing” across tasks, which is the main benefit of meta-learning, in which we aim to learn a unified embedding function across tasks.
>
> Many methods in our survey of literature (methods in the “first camp” in Section 2) indeed use kernels in the context of few-shot meta learning. For example, Matching Net and Relation Net both use similarity kernels. Matching Net relies on a fixed attention kernel (cosine similarity), and Relation Net uses a "relation module" as a trainable kernel that calculates example-to-example similarity scores. DEN outperforms both of these in our experiments.
>
> There are two major reasons that we chose to use neural nets for embedding. First, due to the recent advances in set embedding through deep learning, deep neural nets offer a convenient way to learn the embedding with a variable-length input, which is the main challenge of our setting. Second, in the meta-learning setting, the pretraining data is usually abundant, which allows us to use a more flexible embedding module to improve performance. Our ablation study shown in Table 6 suggests that a reasonably flexible embedding block delivers better performance than a simpler one.
>
> On a side note, Iwata and Kumagai (2020), the method that considers the same heterogeneous distribution tabular data setting, also uses a neural net for embedding, though of a different architecture.
>
> **Re: numerical comparison with non-deep methods**
>
> Thanks for your suggestion. We have performed additional numerical studies on common methods of various complexities trained directly on the small support set of size 50.
>
> On the 20 OpenML binary classification tasks, the average test AUC of a linear model (between 4 and 26 model parameters) trained directly on the support sets is 57.15%; the average test AUC of a 1-hidden-layer, 8 hidden-node DNN model (between 40 and 216 model parameters) is 54.58%. With 2 hidden layers (between 104 and 280 model parameters), the test AUC drops to 53.00%, and with 3 hidden layers (between 168 and 344 model parameters), the test AUC further drops to 49.74%. All these methods significantly underperform DEN, which has an average test AUC of 70.12%. This comparison should demonstrate the benefit of DEN in the small tabular data setting. On the more challenging 8 OpenML multiclass classification tasks, DEN still slightly outperforms the best performing DNN trained directly on the support set (48.60% average multiclass test accuracy vs 45.68%). We will add these comparisons in the revised paper (to be uploaded in a few days).
>
> **Re: adopting the empirical term as an approximation of the ground truth distribution**
>
> In standard few-shot meta-learning, it is often assumed that the tasks are homogeneous, and we observe a handful of labeled examples (e.g., 5 per class) in the support set. In our setting, we allow the tasks to be heterogeneous, which is more challenging than the homogeneous setting. To overcome the challenge, we assume that we have access to a slightly larger support set (e.g., 50 examples in total across all classes) from the target task. Hence (i) the class is not necessarily balanced in the support set and is representative of the test class distribution of the target task, (ii) we have sufficient data to estimate the marginal distribution $P(y)$ (i.e., L-1 parameters for an L-class classification task) using the sample, which is the only model-free empirical estimate we need. We have added a sentence in the introduction to emphasize this distinction.
>
> **Re:  information/analysis about the representation learning**
>
> Visualizing and analyzing the learned representation is helpful, but is incredibly difficult in the meta-learning setting. The embeddings would be dependent on not only the examples, but also tasks. We are not aware of any meta-learning methods that extensively study their learned representations. That being said, we did perform ablation studies on $r$, which suggests that the choice of the embedding block does affect the model performance, and a reasonably flexible embedding module with $r=2$ usually delivers the best performance.
>
> **Reference**
>
> Tomoharu Iwata and Atsutoshi Kumagai. Meta-learning from tasks with heterogeneous attribute spaces. In NeurIPS, 2020.

---

> ### Author Response · Authors · 2022-09-01
> **Response to your more specific questions**
>
> **Re: In Eq(2), the task is represented by the label specific features and label average. While this seems a bit heuristic since the different labels across the tasks do not necessarily have the same label space. E.g, for two different binary classification, the same label average is not meaningful.**
>
> It is true that different labels across the tasks may not live in the same label space, but all of them are encoded as 0 and 1. So the label average always characterizes the marginal distribution of the label. For example, for two tasks $T_1$ and $T_2$. Let $Y_{T_1} \in \{\text{Cat}, \text{Dog}\}$ and $Y_{T_2} \in \{\text{Shirt}, \text{Dress}\}$ be the labels of two tasks. We can encode Cat (Shirt) and Dog (Dress) as $0$ and $1$, respectively. Then the label average of task $T_1$ is an empirical estimate of the marginal probability $P(Y_{T_1} = \text{Dog})$, and similarly for $T_2$. DEN does not need to know the specific label space – it only needs to know the task-specific marginal probabilities for eq. (2).
>
> **Re: In Eq(3,4), the combination of learned embedding Z and task specific embedding S are concatenated, what is the support of combining information in this manner? Why not tensor product?**
>
> Concatenation is the most general way to combine vectors without losing any information. Moreover, Proposition 4.3 shows that, under appropriate conditions, the optimal Bayes classifier (8) can be written as a function of the concatenation of $Z$ and $S$. Inner products are more commonly used to measure the similarity of two tensors, which is a very specific way of combining information. Outer tensor products are not commonly used in embeddings. Moreover, both of these can be represented as a function applied on the concatenation of $Z$ and $S$.
>
> **Re: Besides, why introduce a,b in the embedding S, what is the specific benefit? This makes both high memory and computational complexity. It seems this paper did NOT report the comparison of time and memory complexity.**
>
> Due to the challenge that the input dimension can vary across tasks, the dimension of the embedding $S$ can also vary – think the mean vector (embedding) of Gaussian distributions with different dimensions. To solve this problem, we decompose $S$ into smaller pieces of fixed length $r$ (we use $r = 2$ for illustration and that’s why we introduce a, b). The larger $r$ is, the more expressive the model will be. This point has been explained in the last paragraph on page 4 and Remark 4.1.
>
> Due to parameter sharing in the $h$ and $\varphi$ functions, the decomposition into pairs of inputs $a, b$ does not affect the model size by much. Theoretically, for a DEN model with D inputs, the feature transformation block includes $O(D)$ parameters. For an embedding block with embedding order $r$, $L$ layers and $H$ hidden nodes per layer, the distribution embedding block has $rH + H^2(L-1)$ model parameters. For a classification block with $L$ layers and $H$ hidden nodes per layer for both the $\varphi$ and $\psi$ functions, the classification block has $(r + 2)  H + 2H^2 L $ parameters. Therefore, this DEN model is roughly of the same size as a 3L-layer H-wide DNN model. We will add these computations in the revised paper (to be uploaded in a few days). In our experiments, the model with $r = 2$ outperforms competing methods in many tasks while maintaining comparable training and evaluation time and memory usage.

---

> ### Comment · Reviewer_gQmZ · 2022-09-12
> **Additional Comments**
>
> Thanks for your responses and update paper. I quickly went through them. The followings are my thoughts:
>
> > Hence (i) the class is not necessarily balanced in the support set and is representative of the test class distribution of the target task, (ii) we have sufficient data to estimate the marginal distribution
>
> In few-shot settings with diverse tasks, as far as I understand, the label distribution could not be accurately estimated. The empirical distribution $\hat{P}(y)$ could have a huge gap between the ground truth label distribution. In particular, the tasks are not necessarily related, thus I am not sure when this argument is true. It requires clear and rigorous justifications rather than wording explanations.
>
> > Visualizing and analyzing the learned representation is helpful, but is incredibly difficult in the meta-learning setting. The embeddings would be dependent on not only the examples, but also tasks. We are not aware of any meta-learning methods that extensively study their learned representations. That being said, we did perform ablation studies on , which suggests that the choice of the embedding block does affect the model performance, and a reasonably flexible embedding module with  usually delivers the best performance.
>
> I would say the numerical value is fine but the paper lacks different ways to clearly justify the value of proposed approach. In particular, the proposed approach is a combination of different elements. While in TMLR, the novelty is not critical but it is strongly suggested to illustrate the necessity of each components. Why each component should be considered rather than other alternatives? This could significantly improve the paper.
>
> > Answer 3 Then the label average of task  is an empirical estimate of the marginal probability
>
> This estimation has a significant gap with the ground truth distribution in few-shot settings, where it is consistent with my first concern.
>
>
> Overall, I would really appreciate the effortful works by the authors. The concept within the paper is very important and interesting. While as for the theoretical aspect, I feel quite a bit confused about the analysis within the paper. Without clearly illustrating the assumption and math justification, I am relatively unconvinced about the method within the paper.

---

> > ### Author Response · Authors · 2022-09-13
> > **Response to Your Additional Comments**
> >
> > Thank you for your additional comments! We address your major concerns below.
> >
> > > The empirical distribution $\hat P(y)$ could have a huge gap between the ground truth label distribution.
> >
> > Due to the challenging nature of our setting (i.e., the tasks are not necessarily related), we made an assumption that is **slightly different** from the one in standard few-shot learning: we have access to a small (about 50) support set consisting of **i.i.d. examples** $(X_1, Y_1), \dots, (X_n, Y_n)$  from the **target task**. Suppose that we want to estimate the marginal $p = P(Y = 1)$. Then the sample average $\hat p := \frac1n \sum_{i=1}^n 1\\{Y_i = 1\\}$ is an unbiased estimator of $p$ with standard error $\sqrt{p(1-p)/n}$. For instance, when $n = 50$ and $p = 0.3$, the 90% CI of $\hat p$ is roughly $[0.2, 0.4]$. This means $\hat p$ is an accurate estimate of the marginal $p$.
> >
> > > The proposed approach is a combination of different elements. Why each component should be considered rather than other alternatives?
> >
> > We would like to emphasize that our proposal is one of the first works that address this variable-length input setting. Our major contribution is the novel three-block architecture rather than the individual blocks. We only use simple structures for each block (i.e., PWLs and Deep Sets with MLPs) while still achieving good performance. We have done ablation studies that empirically examined the effect of each block (see the response to reviewer d2m2). We do not claim PWLs and Deep Sets are the optimal layers -- we actually mentioned in the paper that people are free to substitute those layers with other ones that fit into the three-block architecture, such as Set Transformers.

---

### Review · Reviewer_d2m2 · 2022-08-19

**Summary Of Contributions:**

The paper introduces a new Distribution Embedding Networks that focuses on the traditional tasks studied by meta learning: training with certain tasks and then generalize to new ones. Different from the focus studied by most of other meta-learning works, this paper focuses primarily on the tabular data, which might be the reason that this method manages to get a clear improvement over other existing methods.

**Broader Impact Concerns:**

no issues noted.

**Requested Changes:**

Please answer the questions above.

**Strengths And Weaknesses:**

-Strengths
  - The idea of Distribution Embedding Networks is quite interesting, and, to my knowledge, very different from many other meta-learning settings.
  - The empirical performances are extremely strong

-Weakness
  - It's unclear where the power of the empirical performances is derived, some ablation studies are necessary.
  - There seems to be no apparent reason why the study focuses on the tabular data.
  - Proposition 4.3 seems to be a trivial result by reading the proofs in the appendix, also, the authors need to explain the significance of this result, there seems to be no mathematical significance of the result (which is fine), but then, there seems to practical significance either, what does the result guide us in using the method in practice?
  - It's intuitively unconvincing that the method can be applied to arbitrary query tasks with arbitrary support tasks, yet it's not clear where the relation is specified, in other words, what are the application scenarios for this method? This also corresponds to the above question that there seems to be a missing link between the theoretical discussion and the practical side of the work.

---

> ### Author Response · Authors · 2022-08-27
> **One Clarifying question**
>
> Dear Reviewer,
>
> Thanks for your careful reviews. We will post a complete and detailed response to your questions and suggestions in a few days.
>
> In the meantime, to ensure a satisfactory response, we have the following clarifying question:
>
> You suggested us to perform ablation studies to better understand the effect of each block. We indeed performed substantial ablation studies in the paper:
> 1. In Tables 5, 6 and 7, we empirically examined the effect of (fine-tuning) the feature transformation block. The result showed that the transformation block is essential for good performance when the training and test tasks have heterogeneous feature distributions. However, the PLF block is not helpful when the training and test tasks are similar. This is a key finding, which suggests that the feature transformation block is likely the reason of good performance of DEN in OpenML benchmarks -- none of the other methods employ a feature transformation block.
> 2. In Figure 3, we empirically examined the relationship between the performance of DEN and the flexibility of the feature transformation block given different sizes of the support set. The result shows that a more flexible transformation block is helpful with a sufficiently large support size, but could be detrimental when the support set is small. This is intuitive as we need to fine-tune the feature transformation block on the support set.
> 3. In Table 6, we empirically examined the relationship between the performance of DEN and the flexibility of the distribution embedding block. The result shows that the DEN's performance is relatively stable when the embedding block is not excessively flexible. It also shows the importance of decomposing features into pairs in the distribution embedding (as in eq. 2).
>
> Is there any other ablation study that you suggest us to perform?

---

> ### Author Response · Authors · 2022-09-01
> **Response to your questions**
>
> Thank you for your careful review of our paper. Please see our response to your questions below, and let us know if you have further questions. We will upload a revision of this paper in the next few days.
>
> **Re: ablation studies.**
>
> In this paper, we have done extensive ablation studies on the feature transformation and distribution embedding blocks in Section 5. We found that the inclusion of the feature transformation block was likely the reason that DEN outperformed other methods in the setting with heterogeneous task distributions. Please see a detailed discussion in our previous post “One Clarifying question”.
>
> **Re: focus on tabular data.**
>
> Meta-learning with heterogeneous task distributions is an important but under-explored problem due to its challenging nature. As we discussed in the Introduction, tasks with heterogeneous distributions are more prevalent in the tabular data setting (also see our response to your last point), and hence we focus this work primarily on tabular data. The most similar work we are aware of that studies meta-learning with heterogeneous task distributions (Iwata and Kumagai, 2020) also considers the setting with tabular data. We followed their experimental setup for a more fair comparison. Finally, we have suggested directions to extend our work to other types of data (e.g., images) in Section 6 which are interesting venues for future work.
>
> **Re: significance of Proposition 4.3**
>
> We do not claim mathematical significance for this proposition. The significance of this proposition lies in showing that, under appropriate conditions, the model class DEN is expressive enough to include the optimal Bayes classifier. This justifies our choice of the DEN architecture. We have made this point clear in the paragraph below Proposition 4.3.
>
> **Re: application scenarios**
>
> The application scenarios of DEN would be similar to that of Iwata and Kumagai (2020). For example, in numerical studies, DEN shows convincing performance by pretraining them on a few tabular data OpenML tasks, and fine-tuning and applying them on other unrelated tabular data OpenML tasks. In numerical studies, DEN often outperforms other meta-learning methods we compared against (including Iwata and Kumagai, 2020).
>
> Theoretically, Proposition 4.3 characterizes one type of application scenarios for DEN – learning across tasks whose conditional distributions of the PWL-transformed feature $p(z | y = k)$ admits the same $f$-expansion. Heuristically, this means DEN is ideally applied to meta-learning settings in which features across tasks can be transformed to have a similar structure. In contrast, existing meta-learning approaches usually do not have this transformation block and thus require the raw features across tasks to have a similar structure – this is often the case for image data tasks in which features correspond to pixels, but is rarely the case for tabular data tasks. This point was outlined in the paragraph below Eq. (1) and we will emphasize the comparison with existing meta-learning approaches in the revised version (to be uploaded in a few days). In practice, it is admittedly hard/infeasible to check whether the conditions of Proposition 4.3 hold. However, empirical results show that our method delivers good performance overall.
>
> **Reference:**
>
> Tomoharu Iwata and Atsutoshi Kumagai. Meta-learning from tasks with heterogeneous attribute spaces. In NeurIPS, 2020.

---

> > ### Comment · Reviewer_d2m2 · 2022-09-29
> > **Response to rebuttals**
> >
> > Hi,
> >
> > Thank you for these detailed responses, in fact, most of my questions are answered by your answer with
> >
> >    >  Theoretically, Proposition 4.3 characterizes one type of application scenarios for DEN – learning across tasks whose conditional distributions of the PWL-transformed feature ... admits the same f-expansion. Heuristically, this means DEN is ideally applied to meta-learning settings in which features across tasks can be transformed to have a similar structure.
> >
> > Which I think is critical, and didn't really show up in the original writing (or at least in clear places that I noticed), thus I asked for ablation studies (the offered ones do not seem to answer the question of what if the methods are applied to tasks not within the family), theoretical details (where this assumption plays the central role), and application scenarios (how this assumption is represented in the real-world, as you already answered).
> >
> > I will recommend the authors consider clearly stating this application scenario with the details in this response, hopefully in clear places at the introduction, as well as in the form of a remark after the theoretical discussion. In particular, the current writing at the contribution list at the introduction (arguably the most important summary of the paper), the authors have
> >
> >   > We propose a method for few-shot meta-learning with possibly unrelated training tasks
> >
> > which I think can possibly mislead the readers to believe that the method can be applied to arbitrary tasks, which is clearly impossible.
> >
> > Most of my concerns should be resolved with this easy fix. Since the discuss phase has passed and these are easy fixes, I will proceed by assuming the authors will agree to fix these if accepted.

---

### Review · Reviewer_jvsS · 2022-08-24

**Summary Of Contributions:**

This paper proposes a meta-learning framework that can be used with heterogeneous covariate spaces, where the inputs for each task may be incompatible and/or have different dimensionalities. To handle such heterogeneity, the authors have proposed to learn a task-specific linear embedding for each task to send them to a more compatible, continuous space across tasks, while learning a task-independent distribution embedding block to summarize the task distribution and obtain a pairwise embedding of the set of features. Such a treatment of the features as a set allows to deal with variable length inputs that are dimension-wise incompatible. Finally, the authors propose to use a deep-set based classifier to deal with the input feature set. The authors provide theoretical analysis to justify the choice of architectural design, and further perform experiments on a tabular dataset against existing methods, which shows that the proposed framework outperforms them.

Contributions:
- Proposal of a meta-learning framework that can deal with heterogeneous input spaces across tasks.
- Experiments on a tabular data benchmark which shows the effectiveness of the proposed method.

**Requested Changes:**

- Please discuss [Lee et al. 20] and [Lee et al. 21], and modify the hierarchical set encoding in the two previous works into a flat set encoding and compare its performance on the tabular dataset benchmark, if possible.

- If possible, please compare your methods on the any-shot any-way classification benchmark from [Lee et al. 20].

**Strengths And Weaknesses:**

Pros

- The motivation and the focus of the paper, which is to deal with input heterogeneity, is both sound and clear, and to my knowledge, this is the only work on meta-learning that focuses on dealing with heterogeneity in the tabular data.

- The experimental comparison against existing methods as well as ablations of the proposed framework shows the effectiveness of the proposed framework as well as its individual components.

- The theoretical analysis further justifies the choice of the architectural components (e.g. deep sets-based classifier).

- The paper is overall well-written and well-structured, and both the methodology and the experiments section are easy to follow.

Cons

- This is not the first work that deals with varying number of inputs, since [Lee et al. 20] and [Lee et al. 21] use set encoding to deal with datasets with different number and order of classes. Although these previous works do not tackle dealing with tabular data and does not use task-dependent embedding, the strategy used in [Lee et al. 20] and [Lee et al. 21] can be also utilized for tabular data with little modification. Thus, they should be acknowledged and be compared against. Maybe the proposed DEN could be compared against the L2B framework in [Lee et al. 20] in their any-shot, any-way classification setting since the goals for the two works are similar.  The task embedding using set encoding is further developed in [Lee et al. 21], and is used to generalize across datasets with different sizes and orders of classes, although experiments on NAS may be out of scope for this work.


- The baselines used are somewhat dated, although there is one baseline from a work published in 2020. The authors should compare against more recent state-of-the-art meta-learning models from the recent years. Also, they are not direct competitors since they lack mechanisms to handle heterogeneous input distributions.

[Lee et al. 20] Learning to Balance: Bayesian Meta-learning for Imbalanced and Out-of-distribution Tasks, ICLR 2020
[Lee et al. 21] Rapid Neural Architecture Search by Learning to Generate Graphs from Datasets, ICLR 2021

---

> ### Author Response · Authors · 2022-08-27
> **One Clarifying Question**
>
> Dear Reviewer,
>
> Thanks for your careful reviews. We will post a complete and detailed response to your questions and suggestions in a few days.
>
> In the meantime, to ensure a satisfactory response, we have the following clarifying question:
>
> You suggested us to compare and contrast our work with [Lee et al. 20, 21]. We agree that those two papers are relevant and we will discuss them in details in the revised paper. However, although these two papers are designed for any-way (variable number of classes), any-shot (variable number of examples per class) meta-learning, they are not capable of handling variable number of features, which is the main advantage of GON over classical meta-learning methods. To numerically compare DEN with Lee et al. 2020, we will need to modify Lee et al. 2020. One way is to pad the features so that all tasks have the same number of features, same as what we did for other classical meta-learning methods in our numerical studies. In this case the methods are likely to perform poorly. Is there a way that you suggest us to compare with Lee et al. 2020 fairly?

---

> ### Author Response · Authors · 2022-09-02
> **Response to your questions**
>
> Thank you for your careful review of our paper. Please see our response to your questions below, and let us know if you have further questions. We will upload a revision of this paper in the next few days.
>
> **Re: comparison with [Lee et al. 20] and [Lee et al. 21]**
>
> Thank you for suggesting the  references on Lee et al. (2020, 2021). We were not aware of these two papers. After reading, we believe these two papers belong to the “second camp” of optimization-based meta-learning methods and are similar to MAML, BMAML and MT-net. Those methods aim to find a good “starting-point” model which can be easily adapted to new tasks with little fine-tuning. Unlike MAML, which uses a task-independent loss weight and learning rate in finding the “starting-point” model, Lee et al. (2020, 2021) proposed to learn task-specific parameters for the loss weight and learning rate, and thus their methods work better for out-of-distribution tasks than MAML, MT-net etc.
>
> Although DEN and Lee et al. (2020, 2021) are both designed for out-of-distribution tasks, they have a few key differences. First, Lee et al. (2020, 2021) are optimization-based methods, and are more similar to MAML, BMAML and MT-net than to DEN. We have compared DEN with MAML, BMAML and T-net in numerical studies in the tabular data setting, in which DEN delivered substantial performance improvement over these three methods. Second, although Lee et al. (2020, 2021) are designed for any-way (variable number of classes) any-shot (variable number of examples per class) meta-learning, they are not designed to hande variable number of features, which is the main advantage of DEN over existing meta-learning methods. Thus, Lee et al. (2020, 2021) cannot be directly applied in our setting without significant modifications, such as changing the NN based embedding to set embeddings (see Figure 3 of Lee et al. 2020), changing the convolutional layers in the prediction tower to set embedding layers, and/or padding features to equal length.
>
> That being said, although there is significant difference between DEN and Lee et al. (2020, 2021), there are certain architectural components that are shared between them, such as task embedding. We will detail the similarities in the revision, to be uploaded in the next few days.
>
> **Re: comparison with methods with heterogeneous task distributions**
>
> Meta-learning with heterogeneous task distributions (i.e., variable number of features and different feature distributions across tasks) is an important but under-explored problem due to its challenging nature. To the best of our knowledge, Iwata and Kumagai (2020) is the only method that is designed for this setting. We compared against Iwata and Kumagai (2020) in our numerical studies under their setting, in which DEN often outperforms their method, sometimes by a wide margin.
>
> **Reference**
>
> Tomoharu Iwata and Atsutoshi Kumagai. Meta-learning from tasks with heterogeneous attribute spaces. In NeurIPS, 2020.

---

### Author Response · Authors · 2022-09-03
**Revision uploaded**

Dear reviewers,

Thanks for your review. We have uploaded a revision to the paper, which includes your suggestions and clarification on your questions. To help you better see the changes, additions to the text are highlighted in red. We will remove the highlight in the camera ready version.

Thanks again for your careful review and insights!